# Is There a Right to Hope That God Exists? Evil and the Principle of Non-Parity

## Jacqueline Mariña

Philosophy Department, Purdue University, West Lafayette, IN 47907-2098, USA; marinaj@purdue.edu

**Abstract:** In this paper, I respond to James Sterba's recent book '*Is a Good God Logically Possible?*' I show that Sterba concludes that God is not logically possible by ignoring three important issues: (a) the different functions of leeway indeterminism (and the political freedom presupposed by it) and autonomy (the two are very different things, even though both go under the name of freedom), (b) the differences in the conditions of agency in God and in creatures, (there is non-parity in how each must apply the single moral law), and (c) the non-parity between our knowledge and God's. I provide a brief summary of Sterba's arguments, and I develop the following points: 1. Sterba's argument against a Free-Will Defense hinges on his conflation of political freedom and autonomy; 2. Sterba's crucial premise for his argument against soul-making theodicies (namely, that the "Pauline Principle" should be applied univocally across God and creatures) is false; 3. Sterba's arguments against skeptical theism depend on his assumption that our knowledge is comparable to that of God. In each case, Sterba either does not recognize non-parity between God and creatures or does not recognize the difference between the profane (e.g., political matters) and the sacred, (e.g., spiritual matters having to do with the inner nature of the soul's development).

**Keywords:** problem of evil; morality; freedom; autonomy; soul-making; skeptical theism

## 1. Introduction

In this paper, I address James Sterba's question of "whether or not an all-good God who is also presumed to be all-powerful is logically possible given the degree and amount of moral evil that exists in our world" (Sterba 2019, p. 1). When I ask if there is a right to hope that God exists, I am assuming that such a right means a *rational* right. Of course, there is a sense anyone should be free to hope in fantastic and utterly impossible things if they so desire, but such hope in something that cannot possibly exist, something that is logically contradictory, is not a rational hope. The right that I speak of here has to do with the *rationality* of hope, and this includes both the logical possibility of its object as well as its motivation. The hope that God exists cannot possibly be rational if God is impossible, nor can it be rational if this hope is grounded in base and unworthy motives. Importantly, the issue at stake is not whether God's existence can be proved, or whether or not there is evidence for or against God's actual existence. The issue is rather whether such a hope is, as J. L. Mackie put it, "positively irrational" (Mackie 1955, p. 200), so that there is no way that one could *possibly* reconcile the idea of a good, omnipotent God with the reality of evil.[1] The problem was already identified by Epicurus: "Is he willing to prevent evil, but not able? Then his is impotent. Is he able, but not willing? Then he is malevolent. Is he both willing and able? Whence then is evil?" The problem, however, is not just that there are evils. It is that there are *horrendous* evils, evils that threaten the very possibility that one's life could be "a great good to one on the whole" (Adams 1989, p. 299). These are evils, not only that one might suffer, but that one might perpetrate. Examples of the former would be being tortured to the point of the disintegration of the personality and then killed or being a mother in a concentration camp forced to choose amongst her children, thereby becoming the agent of evil against a person one loves more than one's own life. Examples

of the latter would be Medea, who in a fit of passion kills her own children,[2] or a torturer that tortures and kills the person he loves the most in a fit of rage. And on a larger scale, there is the Holocaust and Hiroshima and Nagasaki. There are many other bone-chilling scenarios; these are just a few.

Attempting to provide a defense of God when confronting such evils threatens to put one in the position of at least being in some significant sense tone-deaf, or of lacking empathy, or of a certain cold-heartedness. For certainly the appropriate response to such horrors is being struck dumb. It is just too hard to imagine that a good God could be the author of such a world. In providing either a theodicy or defense of God in the face of such evils, one too easily becomes like Job's friends, who did not speak "what is right" about God (Job 42:7). On the one hand, one cannot speak properly of God in such conditions by defending God. For it might seem that one is justifying or defending God against the victim, who like Job remonstrates against God. Such a justification puts one in a position one cannot legitimately occupy, a position "above" the sufferer, in such a way that defending a God who would allow it also seems to imply a justification of the suffering. As if one could fathom why this evil or suffering occurs, or as if one could oneself avoid such a fate by defending God so that one could remain untouched by the evil that in the end consumes all earthly life. This is the position of Job's friends: they argue Job must have done something wrong. On the other hand, one can legitimately take the side of those who suffer evil by validating the legitimacy of hope. For without such hope there is only the possibility of utter despair. And it is in this spirit that I undertake to answer Sterba's charge that a good God is not logically possible.

Sterba attempts to show that there can be no possible morally sufficient reason for God to permit evil. Morally sufficient reasons can be spelled out in two ways. First, if "instances of suffering result *from* goods which outweigh the negative value of suffering" (Pike [1963] 1990, p. 44). Second, if "suffering results *in* goods that which outweigh the negative value of suffering" (Pike [1963] 1990, p. 43). The claim that suffering is a consequence of God granting human beings the freedom to choose between alternatives would be an example of the first. The claim that allowing suffering is necessary for soul-making is an example of the second, Significantly, there are tight connections between freedom and soul-making, but it is important to understand which kind of freedom is at issue, and what those connections are. Sterba does not give a coherent account of what they are. How the connections are envisioned depends on the kind of freedom necessary for soul-making. Sterba argues that soul-making depends on leeway indeterminism, itself depending on there being a kind of outward, or political freedom (if another constrains me, I have also lost my freedom of choice). But, I will argue, the freedom that is important for soul-making is not leeway indeterminism but the capacity for autonomy, that is, the capacity to value what is right and good above all else. This capacity can never be threatened. Furthermore, as I argue below, it may be the case that its development may require many false starts and wrong turns (suffering results *from* those), as well as suffering itself (suffering results *in* purification). Lastly, it is important to note that no matter how many reasons failing to justify the permission of suffering are enumerated, one can never claim to have examined all the possibilities.

According to Sterba, freedom cannot justify suffering, since allowing one person the freedom to significantly harm another limits that other person's freedom. Allowing horrendous evils, supposedly a consequence of the granting of freedom, makes the granting of freedom to everyone impossible, since those who suffer them lose their freedom. Freedom, then cannot provide God with a morally sufficient reason for allowing them. Sterba then argues that horrendous evils also make soul-making impossible, since those who suffer them lose their freedom, hence their opportunities to make choices, and hence for soul-making. Sterba, however, achieves these results by ignoring (a) the differences and relations between political freedom and autonomy (the two are very different things, even though both go under the name of freedom), (b) the differences between the conditions of agency in God and in creatures, implying that there is non-parity in how each must apply the

single moral law, and (c) the non-parity between our knowledge and God's. My response will be divided into four parts. In the first, I provide a brief account of what I think are the fundamental premises grounding Sterba's argument. In the second I look at his argument against the Free-Will Defense. Sterba's argument against the Free-Will Defense depends on his ignoring the idea of freedom as autonomy, which does not depend upon leeway indeterminism. Importantly, soul-making depends on the capacity for autonomy, not on leeway indeterminism or political freedom. Yet, because Sterba has ignored the concept of autonomy, he assumes that leeway indeterminism and political freedom must obtain if soul-making is to be possible. He thereby confuses the functions of indeterminism and political freedom with that of the capacity for autonomy. The third section looks at Sterba's objections to a soul-making theodicy. Sterba's argument depends on his crucial premise that "the Pauline Principle" should be applied univocally across God and creatures. I show that it cannot. Lastly, I look at Sterba's arguments against skeptical theism, which once again depend on his assumption that our knowledge is comparable to that of God's. In each case, Sterba either does not recognize non-parity between God and creatures or does not recognize the differences between the profane (e.g., political matters) and the sacred, (e.g., spiritual matters having to do with the inner nature of the soul's development).

## 2. Sterba's Argument

The argument has many moving parts, but central to its development is what Sterba calls "the Pauline Principle," namely, "never to do evil so that good may come of it." This is supposed to be the central proposition grounding what counts as moral goodness, and hence it is taken as a principle that a good God must adhere to. Sterba takes this as a fundamental moral principal binding both God and human beings. His account greatly depends on a univocal understanding of how it applies to both human beings and God, so that Sterba argues seamlessly from human cases to how the principle should constrain divine action.

Sterba never really gives a fully adequate account of what he means by this principle up-front, and its lack of determination allows him to let it do a lot more work than it reasonably should be allowed to do. On its face, the principle is absurd even when applied to human conduct, since it can, for instance, be taken as grounding views that deny a woman required medical care in a pregnancy emergency if their embryos or fetuses still have a "heartbeat." Women in Texas and other places stand in grave risk of losing their uterus, limbs, mental functions and even their lives since doctors now will not risk helping them until the fetal heartbeat stops. The risk that both will die does not matter, because providing an abortion would be "doing evil" so that "good may come of it." Better, as many on the right have argued, not to intervene at all and let God take care of it.[3] My point here is that to avoid absurdity, the real content of the principle needs to be carefully delineated. We need to ask, for example, what is meant by "evil" and what is meant by "good." Are the good and evil that Sterba is talking about to be understood in terms of pleasure and happiness so that what he means is that we should never cause pain so that another pleasure may ensue? Or perhaps that we should never limit another person's happiness in order that greater happiness is to come of it later? What about the child that needs a painful operation to save their life? Should operating the child without their consent be prohibited? Let us eliminate this option as too simplistic. Perhaps what Sterba means by this principle is just a kind of anti-consequentialism, and in its place, he wants a kind of Kantian theory, where persons should never be used as *mere* means, and their autonomy and personhood must be respected. In this case, the principle would mean something like: never use a person as a *mere* means in order to create greater happiness or well-being, or even greater virtue. In this regard, it is important to stress that Kant held that we use each other as means all the time, and indeed must, given our individual finitude. What is impermissible is taking a person as a *mere* means.

Let us assume that what Sterba really means is the Kantian point that persons have a fundamental value; they are, in fact the ground of all value.[4] From this Sterba infers that

such a value guarantees *rights* or claims, certain things that are due to an agent just in virtue of their personhood. Persons have ends of their own. Using them as a mere means violates the very ground of value (their personhood) by treating them as less valuable than some end independent of this personhood itself. Respecting personhood thereby means respecting both the freedom of each individual and their capacity to progress in virtue, that is, ensuring that they have opportunities for soul-making. Given these inferences, Sterba comes up with the following requirements:

1.  Moral Evil Prevention Requirement 1: Prevent, rather than permit, significant and especially horrendous evil consequences of immoral actions without violating anyone's rights (a good to which we have a right) when that can easily be done.
2.  Moral Evil Prevention Requirement 2: Do not permit, rather than prevent, significant and especially horrendous evil consequences of immoral actions simply to provide other rational beings with goods they would morally not prefer to have.
3.  Moral Prevention Requirement 3: Do not permit, rather than prevent, significant and especially horrendous evil consequences of immoral actions on would-be victims (which would violate their rights) in order to provide them with goods to which they do not have a right, when there are countless morally unobjectionable ways of providing these goods. (Sterba 2019, pp. 126–28)

Based on these principles, along with some other important assumptions, Sterba argues that a good God could not possibly have a justification for allowing horrendous evils. No cogent account of their permission being the only means to assure something of greater value succeeds. First, the Free-Will Defense will not do, since it entails that God allows one person's misuse of freedom to severely curtail another person's freedom. In the case of the victim, God has not preserved *their* freedom. The freedom to commit horrendous evils does not preserve the significant freedom of everyone but only allows the strong to overwhelm the weak, so the latter lose their freedom. If the goal of permitting horrendous evils is preserving freedom, that goal cannot in principle be achieved by such means. An all-good and powerful God would certainly have understood this to be the case. Freedom, then, cannot be the greater good achieved through allowing evils. Surely, argues Sterba, would it not have been better if *everyone's* significant freedom had been guaranteed, so that those with evil intentions would be free, but just not so free as to have the power to make their victims suffer horrendous evils? An omnipotent God certainly could have achieved this. God, for instance, could step in at the last moment and prevent horrendous evils from happening, just like a kind of superman.

Second, the Soul-Making Defense stands or falls with the Free-Will Defense. If freedom is required for soul-making, but the freedom of some to commit horrendous evils compromises the freedom of others, then the latter group will not have the requisite soul-making opportunities if horrendous evils are permitted. The Soul-making Defense, depending as it does on the Free-Will Defense, also cannot justify the permission of horrendous evils.

Third, the permission of horrendous evils for the purposes of soul-making is also not justified if the ends are the provision of goods that have not been agreed to by the persons who participate in them (e.g., the opportunity for forgiveness), or for the provision of goods to which persons do not have a right. For instance, it is certainly reprehensible to think that God allows some individuals to perpetrate horrendous evils on their victims just so that they (the elect) could enjoy divine forgiveness.[5] In the latter case, one would be depriving one person of a good to which they have a right to provide another person with a good to which they do not have a right. That amounts to the mere use of one person for the purposes of conferring a non-merited good on another; in this case, the first person is treated as nothing but a tool. While there may be other details of Sterba's argument I do not mention above, this constitutes the heart of the argument. Some of those details will be discussed in the course of my larger argument.

### 3. Free Will and the Non-Parity between Autonomy and Political Freedom

There are many understandings of what free will amounts to. Sterba does not do enough to disambiguate between them. As I will argue below, Sterba's argument ultimately depends on conflating leeway indeterminism, itself depending on political freedom, with autonomy. He believes that the kind of freedom required for soul-making is contra-causal freedom, so that a lack of political freedom ("the freedom as noninterference cherished by political libertarians") (Sterba 2019, p. 27) limits it, too. He fails to see that the kind of freedom required for soul-making is *not* a freedom of indifference regarding choice, but is, rather, the capacity for autonomy and its development. Insofar as he fails to see this, he treats them as the same kind of things, having the same kinds of conditions for their exercise. They do not. In order to show why this is the case, we need to explore different senses of freedom and how they relate to autonomy and political freedom. I first discuss the original analysis of freedom Sterba works with, one grounding his whole discussion, namely a kind of leeway indeterminism where choice is indeterminate; from this he concludes that without outer freedom, one cannot have the freedom to make choices either, since in such a case one is either constrained, or is robbed of all opportunities to make choices (if, for instance, one is killed). As Sterba notes, "contra-causal freedom presupposes freedom as non-interference: you cannot be contra-causally free to do X if you are interfered with such that you are kept from doing X" (Sterba 2019, p. 27). I show that this understanding of freedom is a red herring since it contradicts the very possibility of having a will. Neither it, nor Plantinga's defense of it is relevant to the sort of freedom that genuinely matters in this context, namely, freedom as autonomy. The autonomous agent is one that acts from the right set of motives, namely, they can love the good for its own sake (Plato) or act for the sake of the moral law (Kant). I take both philosophers as aiming at the same general idea, namely, the autonomous agent understands what is right and good, acts accordingly *because* it is right and good, and is willing to give up the satisfaction of all other desires if morality requires it.[6] Freedom as autonomy is a central notion in the Western philosophical tradition, and is often confused with other kinds of freedom. I argue that only this kind of freedom is necessary for soul-making, and that it cannot be threatened by horrendous evils.

A common understanding of freedom is the capacity to genuinely choose between alternatives, so that at a given moment there is a real possibility that the agent can do a or b or c (or whatever number of real alternatives there might be in the case at hand). By this real possibility, we do not just mean that the conditions for freedom of choice can be met in this way: *if* an agent had a different desire, then they would have chosen differently. In these cases, all that is meant is *if* there had been a different causal chain, then things would have turned out differently. But that does not mean that given a particular causal history, an agent can do *either* a or b. It is the latter that we require for this real possibility of choice: same individual, same past, same laws in play, and yet different possible outcomes at the moment of action.[7] Following Pereboom, let us call this leeway indeterminism. I do not think that such an understanding of free will is internally coherent.

Even putting aside the question of causal chains, the only way we can make sense of choices is in terms of motives. The agent must have some kind of reason or end in view for their choice to make any sense. That doing a particular action was something that they found valuable at a given time can only be explained in terms of their prior beliefs and values, in short, in terms of their character. As Hume and later Mackie (1955, p. 14), would point out, short of such an account, where the action can be linked with character, with what the agent finds to be of value, the action winds up being a random one. And if the action cannot be linked with a person's character, it is hardly attributable to them. Hume put the point nicely when he noted that if we deny "necessity," the individual is "as pure and untainted, after having committed the most horrid crime, as at the first moment of his birth, nor is his character anywise concerned in his actions, since they are not derived from it, and the wickedness of the one cannot be used as proof of the depravity of the other" (Hume [1777] 1975, p. 98). A coherent theory of action requires, at the very least, that the action can be joined to the character, so that it can be understood in terms of a

person's wants, wishes, desires, and beliefs. It must be possible to give some kind of an account of *why* a person embarked upon a particular course of action if we are to attribute that action to them, that is, we need to be able to understand an action in terms of what a person considers valuable or *worth* doing. We cannot sever the relation between action and character without making the very possibility of having a will (where action is intentional and not random), incoherent. And even supposing that freedom consists in choosing a character (so that the actions then flow from the freely chosen character), given the lack of a basis establishing what is of value to the agent, that choice of character would have to be a random one. Actions taken for no rhyme or reason are random actions, and random activity stands in contradiction with the very notion of a will.

For these reasons I do not think that Plantinga's particular brand of a free-will defense is plausible, so I will not try and defend it here.[8] But this seems to imply that if God produces agents with given natures or characters, and those must work themselves out in given ways, then Mackie is correct. At first blush at least, it seems it should be possible that God could have created creatures that he foreknew would always choose the good freely,[9] for in creating them he would have been fully aware of their nature, and hence what they would have considered valuable at each point in their development. Those creatures would have been familiar with the ultimate good and its value; they would have tasted it, and *nothing* would or could ever lead them to deviate from it. Importantly, this is the final state of bliss that Christians hope to achieve; otherwise, the whole painful drama of redemption through which they came to know the depths and expanse of God's love would still not be enough to keep them from randomly turning away from God at any moment, so that the whole of human history would have been for naught. But if it is the case that the creature can indeed reach a state of knowledge or participation in the Good such that they will never turn from it or from God's love, then the question remains: why did God not make those creatures that way to begin with? A great deal turns on whether, in fact, it would have been logically possible for God to create those kinds of beings.

Importantly, a soul-making theodicy does require that we attribute a particular kind of freedom to the creature. It is not leeway indeterminism. Rather, on this account, an individual would *not* be free if it were the case that their only capacities were to love God, Justice, or the Good out of a fear of hell, or for a desire for, as Sterba puts it, "a consumer good" (Sterba 2019, p. 37). Rather, each individual must be able to develop into the sort of being that can love the Good for its own sake, and not just for its consequences. Think of Glaucon's challenge in book two of the *Republic.* We test whether a person loves justice for its own sake by stripping them of all the *consequences* of justice. That person winds up with a lifelong reputation for injustice, all earthly goods are taken from them, and finally, they wind up on a rack, their eyes gouged out. A person *must be able* to undergo all of this for the sake of justice if we are to be able to say of them that they loved justice, or the Good, for its own sake. Kant makes the point much more precisely: the good person makes action in accordance with the moral law the condition of the pursuit of happiness, the implication being that they must be able even to give up their life if moral action required it. Their soul must be ordered in such a way that they can do this, that they can recognize what is right and good as a transcendent value outweighing all earthly goods. Becoming that sort of being may require an enormous amount of moral development which may involve a great deal of suffering.

This is freedom as a kind of *autonomy*. Importantly, it does not require leeway indeterminism, namely, the capacity to choose a or b given the same set of conditions. It may be the case that all creatures are destined to develop into the sorts of beings that can love the good above all else, and that each creature has a specific path that it must traverse to get to that point. This kind of freedom may require, however, that we think of the individual as the ultimate *source* of their actions, so they are not impelled by causal factors outside their will: we must be able to say it is *they* who are developing, and who are playing an active role in their own development. An individual cannot be understood as a mere segment of a causal chain without any agency of its own. But being the source of your own action does

not imply leeway indeterminism; in fact, if actions are not determined by what a person values (if they were random), then it would be hard to attribute the action to the person in the first place.

Importantly, if freedom is understood in this way, as a kind of autonomy, then it is not the case that when one person limits the bodily freedom of another, even by causing their death, they eliminate that person's autonomy. Sterba's argument only seems to work because he conflates two kinds of freedom: freedom to do things in the world, and freedom as autonomy. He claims God's permitting evil is not justified by the opportunities for soul-making it provides

> if having opportunities for soul-making in our world is dependent on having significant freedom such that a net loss of significant freedom in our world would result in a net loss of the opportunity for significant soul-making as well. (Sterba 2019, p. 35)

Sterba's confusion, then, is that he believes a kind of political freedom is necessary for soul-making. Based on this confusion, he concludes "accordingly, both God and a just political state should be focused on preventing the significant consequences of moral evils, making no attempt to prohibit all moral evil because that would interfere with a person's significant freedom" (Sterba 2019, p. 59). If outer freedom were necessary for soul-making, or if the limitation of a person's outer freedom would necessarily result in a loss of autonomy in the sense defined above, then Sterba's argument would go through. But the freedom to act in the outer world (which is the kind of freedom we are talking about when we talk about political freedom) is of a very different sort than inner freedom, which concerns *what is valued* and the motives for caring for it. It is certainly possible that no external circumstances can compel an individual to have certain values. And if this is the case, the freedom of one person to inflict horrendous evils on one individual (Glaucon's torturer) does not limit the inner freedom of the individual who loves or is learning to love the good above all else. In fact, for all we know the test of such an experience might even be an important one in a soul's development, for through it the soul comes to know itself in a certain way, as capable of valuing certain things above all others. Since it is the capacity for the latter kind of freedom (freedom as autonomy) that is required for soul-making, there is no contradiction in the permission of horrendous evils and the supposition that *everyone* can maintain freedom as autonomy, or can exercise this capacity even if it is still in development. This kind of freedom just cannot be taken away. Hence, Sterba's claim that "the freedoms that victims lose by the serious wrongdoings of others are much more important than the freedoms that are exercised by those who wrong them" (Sterba 2019, p. 53) depends for its plausibility on a somewhat superficial understanding of freedom. Inner freedom, autonomy, cannot be taken away through outer actions. Replying to this objection, Sterba notes, "Nor would it do to claim that the freedom that is at issue here is an inner freedom of the will that could not be affected at all by external circumstances. This is because if that were the only freedom that was at issue here, God would have prevented all the evil in the world without interfering with this freedom at all" (Sterba 2019, p. 27). This account ignores the possibility that suffering may be something the soul must undergo for soul-making: the development of this inner kind of freedom of the soul, where the self discovers what its "proper self" values, may require not only many false starts, but many trials and tribulations as well.

Further, if we admit to the doctrine of reincarnation, even if one individual does terrible things to another, there will be multiple opportunities for both to finally get things right, for the torturer to make up for what they have done, and for the victim to continue their progress in virtue in whatever way is necessary. There is no contradiction in supposing that the experiences of all creatures can be harmonized in such a way that through their actions they each become the means for the moral development of the other.

## 4. Soul-Making & the Non-Parity of the Application of Moral Requirements

Sterba's argument against soul-making theodicies comes on the heels of his argument against the freedom defense. He notes that evil cannot be justified by soul-making "if having the opportunity for significant soul-making in our world is dependent upon significant freedom such that a net loss of significant freedom in our world would result in a net loss of significant soul-making as well" (Sterba 2019, p. 35). Sterba claims that soul-making *is* dependent on our having significant outer freedom in our world. Importantly, to make this sort of claim Sterba needs to know just what soul-making really amounts to and the kind of freedom needed for it to happen. Yet, he seems to assume to know what it is, and that the "natural" opportunity for soul-making requires people to have bourgeois lives where nothing terrible ever happens (Sterba 2019, p. 84). In such a secure environment, people can develop as honest, responsible people that love their family and their work, and that should be enough for them to enter the kingdom of heaven. But what if genuine soul-making requires a lot more? What if the important freedom at issue is not a kind of political freedom in which you are free to pursue happiness and "bourgeois" opportunity for soul-making, but the development of autonomy? What if what is required is that the soul come to know itself as loving justice (or God, or the Good) above all else, and what if this just really takes a lot? Even the terrible Nietzsche claimed that it was through the hostility and cruelty of the human being's instincts turned backwards against itself, in short, through suffering, that the soul developed.[10]

Sterba, however, argues that it *should* not be the case that soul-making requires suffering, or at the very least, not too much of it. If it did, then God would be justified in allowing for or, even ultimately, arranging for terrible things to happen to people so that their souls can develop. But this, argues Sterba, would make God complicit in immorality. Sterba offers the example of parents permitting their children to be brutally assaulted for soul-making opportunities: they would have the opportunity to forgive, their comforters to comfort, and their tormentors to repent and be forgiven (Sterba 2019, p. 57). That, of course, would be reprehensible. It would also be reprehensible for any one of us to make another suffer because we think that they are immoral, they deserve it, or they need a bit of an opportunity for soul development.

This is the crucial premise driving Sterba's argument: "*If it is always wrong for us to do actions of a certain sort, then it should always be wrong for God to do them as well.*" (Sterba 2019, p. 57). In adjudicating this claim, two issues must be considered.

The first is that there is a single moral law. If we are to think of God as good, the *same* standards of goodness must apply to both God and creature. There is certainly something wrong with the claim that God is bound to moral standards different from our own, or that whatever God wills is what turns out to be good (a kind of voluntarism). Whatever idea we have of God, irrespective of its source, we must first always ask ourselves whether the God we imagine is worthy of worship, and if so, then such a God must conform to moral concepts.[11] These cannot themselves be derived from our idea of God, since it is we ourselves that must compare that idea with moral concepts to judge whether such a being is worthy of our worship to begin with.[12] I have already noted the inherent lack of clarity in what Sterba's Pauline principle enjoins. For our purposes, we can begin with the intuitive Kantian principle that persons should not be treated as a *mere* means to achieve another person's ends; each person has an inherent and absolute value. On this account, a God that creates beings predestined to eternal damnation and suffering would not be worthy of worship. That would be a clear example of a violation of the inherent worth of persons.

The second concerns the *way* in which this fundamental moral principle is applied. And it is here that Sterba goes radically wrong. For granted there is a single moral law through which the good is established, and so determinative of how we must think of an omnibenevolent will, it is not the case that this single moral law can be applied *univocally*. The conditions of agency dictate how it is to be applied and have a decisive influence on whether an action turns out to be wrong. These conditions are significantly equivocal across God and creatures, and so there is non-parity regarding how moral requirements apply to

us and apply to God. We have limited knowledge, capacities, and love. The conditions of finitude severely restrict what is morally permitted to us. For instance, it would be wrong to for us to personally "punish" a wrongdoer or to try and give them what we consider their just deserts. There are certainly moral limits to how much each of us can interfere with other people's free choices, for instance, those of our grown children, even when we think they are making terrible mistakes. There are numerous reasons for these limits having to do with our own very own limited development. First, we do not know other people's hearts; only God can judge the heart. This ignorance of the inner states of others also means we do not know what will ultimately fulfill them. That is something that we must allow them to find out for themselves. Second, even when we wish the absolute best for another, we have little control over what ultimately will befall them; a parent, for instance, may force a child to go into a particular field "for their own good" that ultimately leads to their ruin, a ruin the parents were powerless to prevent. Third, and perhaps most importantly, we are limited in love. For instance, too often our desire to punish is just vindictiveness or is at the very least tinged with it. And too often our belief that we know what is better for the other is just a façade for our desire to control and to have power over others. Such dynamics often prevail in family life. Lastly, we have little or no understanding of the final *telos* of the soul's life; having no such understanding of it, we are hardly in a position to interfere too much in helping the other achieve it.[13]

These conditions would not hold for God, who is perfect in wisdom, power, and love. God knows the heart's fundamental desire and what is necessary for the individual to achieve it. God knows the condition of a person's soul at every given moment of their life, and which elements of the soul may need correction if the individual is to be able to fully understand and participate in what is of true worth. Having perfect love, God fully wills the complete fulfillment of each person, (in the Christian tradition this ultimately means being taken up into the divine life itself, that is, to become God with God). And finally, having perfect power God can ensure that this end will be achieved. For these reasons, while there is a single moral principle, what it means for God to adhere to it amounts to something very different from what it means for a creature to do so.

It is reprehensible for a parent to inflict suffering on their child so that child can have soul-making opportunities because the parent is not God, the parent does not know very much at all about the inner development of the child's soul; they do not know what stage it is at and could not possibly know what the child really needs in relation to that spiritual development. The same is true in cases where one individual thinks they should "punish" another. No one has a right to punish the other because they do not know the heart of the other, and they never fully love the other, either. These limitations do not hold of God, who has both perfect wisdom and love. If God creates creatures whose destiny it is to become God with God, then God is still a good God when God allows horrendous evils, especially if the undergoing of such horrendous evils are necessary experiences through which the soul is prepared to enter the divine life. There may be certain virtues that can be achieved *only* in and through the suffering of certain things so that the suffering is integral to the acquisition of the virtue. God is in a position to know the ultimate needs of the soul, to love it with God's infinite love, and to ensure that the soul arrives at its ultimate destination. Because God is working from these conditions, God's actions are such that the personhood of the creature is respected when God allows the creature to suffer. Further, this personhood is respected if, once the creature enters its full spiritual maturity, it comes to recognize the need to have had to undergo the terrible sufferings that it underwent. Such sufferings would be considered nothing in relation to the ultimate good of entering the divine life. This cuts against Sterba's claim that "victims may never be able to give their informed consent to or find reasonably acceptable the infliction of such consequences [horrendous evils] on themselves" (Sterba 2019, p. 75). Importantly, if this statement concerns a mere possibility, that possibility, which may not be realized, cannot speak against the possibility of God. And if Sterba is making the stronger claim that victims *will* never give this consent, there is simply no way he could know this. Sterba continuously makes claims to which he

is not entitled. From the ultimate vantage point of its full spiritual maturity, it may well be the case that what seemed very real to the soul in its earthly condition was nothing more than a kind of dream. Since God would know the creature in its final perfection, the consent of the fully perfected creature would be assured. A two-year-old may think the discipline and limitations it undergoes a terrible thing, but upon reaching adulthood may come to recognize the need for them. The same holds true for the immature soul, which upon perfection will have a very different view than it had in its immaturity. While we do not know that this story is true, it is certainly possible that something very much like it is.

*Skeptical Theism: Non-Parity between Our Knowledge and God's*

　　Much of what I have argued above has straightforward implications for skeptical theism. The conditions of our knowledge are radically different from the conditions of God's knowledge. We are simply in no position to know what God knows, or to love as God loves. Hence, when Sterba claims he can distinguish between "natural opportunities" for soul-making, and "Godly" opportunities for soul-making, he asserts he knows much more than he possibly can. According to Sterba, "natural" opportunities for soul-making are "the opportunities each of us must have in order to become a good and just human being" (Sterba 2019, p. 83). But what does that really amount to, when we are speaking of the heart that no one but God knows? Or whether "natural opportunities," which do not include horrendous evils, are sufficient to mold the person into the kind of being capable of knowing God? Can we even distinguish between "natural" and "Godly" opportunities for soul-making? Perhaps *all* the moments in our lives are "Godly" opportunities for soul-making, only of different kinds. Furthermore, how does Sterba know what the final telos of soul-making even is, and what it would even take to get there? He assumes that the final *telos* of soul-making is a kind of bourgeois state of happiness, a kind of "consumer good" of which one can become worthy through a bourgeois life, with bourgeois challenges that are not too difficult.[14] But how does he know this? What if the goal of soul-making is so great that we can barely glimpse an idea of it?[15] And what if the path to get there must seem terrible to us? These are things of which we have no knowledge, but they are certainly *possible,* and religious thinkers and mystics report that something like this is the case.[16]

　　The main claims through which Bergmann (2009, 2012, 2014) develops skeptical theism are surely correct: we may not be familiar with all possible goods, and we may not be familiar with all possible entailment relations between all possible goods and all possible evils. Without this knowledge, there is no basis for the kinds of claims Sterba makes, which crucially depend on his catalog of possible goods and possible evils and their possible relation to one another being an exhaustive one. Importantly, the relevance of Sterba's arguments against Bergmann is limited to the relation between events within natural causal chains: Sterba argues that since God is omnipotent and master of all the natural world, it makes no sense to argue that God allows one horrendous evil in order to mitigate the causal consequences of something worse occurring down the line had it not been allowed to occur. God should be able to intervene at *any* moment in a causal chain, preventing both the initial horrendous evil and the possible consequences of its non-occurrence (Sterba 2019, p. 79). The problem here is that Sterba's understanding of the relation between possible evils and possible goods is in general limited to events in causal chains that are experienced by us as good or evil. His discussion of laws of spiritual development is minimal and perfunctory. For instance, he claims to know the following:

> ... it would not be morally appropriate for God to make the provision of a Godly opportunity for soul-making to which we do not have a right dependent on his permitting significant and especially horrendous evil consequences of immoral actions, especially given that a Godly opportunity for soul-making could be provided to us in countless other ways that are morally unobjectionable. (Sterba 2019, p. 95)

But how could Sterba possibly know *which* Godly opportunities for soul-making a soul needs in order to achieve blessedness? Laws of spiritual development can at best be intimated by us "in a glass, darkly," (I Corinthians 13:12) and Sterba can hardly claim that he fully understands how all the experiences in a human life, or assuming reincarnation, all the experiences in many lives, might shape spiritual development. Sterba would not be in a position to assess the nature of ultimate reality or the sacred, or what it would take for an individual to partake of it. He cannot rule out the possibility that there are certain experiences, perhaps absolutely terrible ones, that the soul must undergo in order to be shaped in a certain way, that is, in order for it to be able to come to know and desire the ultimate good. Such experiences might be integral in shaping the soul's very structure of desire.

In the first part of this paper, I suggested that the key issue required in answering Mackie has to do with whether it would have been possible for God to create creatures such that they always freely choose the good. Part of the answer turns on what it means to "freely" choose the good. I have argued that we must understand choice in terms of what an agent *values*, and if this is the case choices cannot be divorced from an agent's character. It makes no sense to speak as if an agent could really have chosen a *or* b at a particular moment, as if the two alternatives could ever appear to them as two equally weighted, live options. The agent will always assess one option as preferable to the other, and this assessment will be determined by their level of experience, maturity, and insight. Could God have created creatures capable of valuing and experiencing Godself without those creatures having had a long history of development through which they came to understand the good?[17] What if the creature must first explore the avenues leading away from God in order to really understand the one that leads to God? What if knowing God implies a certain capacity for *experiencing* God, one that cannot possibly be gained except through a long history of experience through which the soul is shaped? The soul comes to know where certain paths will lead because it has already traversed them, experienced dead ends, and then changed course. This is how it gains the insight necessary for it to participate in the divine life, and how it gains the maturity needed for it to be capable of loving the Good for its own sake, in such a way that once it understands it, *nothing* would ever tempt the soul to do evil or deviate from it. If a soul can participate in the divine life only insofar as it has been shaped, or has gained insight through its history, God cannot just simply create a fully perfected being. For that would mean that God would have to create a being already in possession of the experiential knowledge it requires to know God, without that soul actually having experienced that history. But this is contradictory. To claim that God is not logically possible because he cannot create such a being would be akin to claiming that God is not logically possible because God cannot create a square circle.

At the beginning of this essay, I noted the inherent difficulties of dealing with this topic–arguments must be undertaken in the proper spirit. At this very moment, there are many people experiencing horrendous evils. One need only think of flood victims in Pakistan, victims of starvation in Africa, or victims of war and horrendous torture in Ukraine. These are terrible things, and it is *our* obligation to mitigate suffering. Yet, we would be doing victims no favors in saying to them that their experiences are proof positive that there is no God, that they don't mean anything, won't amount to much in the grander scheme of things, and that their suffering is senseless since all that awaits is the silence of the tomb. Because we are moral beings, the very experience of suffering and evil demands of us that we *hope* that all suffering will be redeemed. The Christian might say that we must hope that the *very terrors* of the cross must be redeemed, that is, that our putting on of Christ and our suffering with him *itself* is shown to have meaning and is not just a means to an end independent of this suffering itself. The soul is made God-like through the experience itself. That God is with the soul as it suffers through this process is a comfort, as is the hope of the future estate achieved through it.

**Funding:** This research received no external funding.

**Acknowledgments:** I would like to thank Paul Draper and two anonymous reviewers for comments on earlier drafts of this paper.

**Conflicts of Interest:** The author declares no conflict of interest.

## Notes

1. As Nelson (Pike [1963] 1990) has clearly demonstrated, the "contradiction" between the proposition that an omnipotent, omniscient, and morally perfect being exists, and the actuality of evil only holds if it can be established that God could not *possibly* have a morally sufficient reason for creating a world in which suffering obtains. Proving that there is no *possible* morally sufficient reason, is however, a herculean task, one that Sterba believes he has achieved.

2. While this is a literary example, there are actual cases such as these.

3. The Idaho Republican Party, for instance, removed the exception for the life of the mother from its platform last July. Scott Herndon, an Idaho Republican running unopposed for a state senate seat, noted "Doctors may not intentionally kill the child in their medical efforts to treat the mother." John Seago, president of Texas right to life, argued that doctors cannot decide that "I want to cause the death of a child today because I believe they are going to pass away eventually" (Goldberg 2022). Mary Siegler, Professor at the University of California, Davis School of Law and an expert on the history of abortion law in the U.S. sums up the reasoning behind these ideas succinctly when she notes, "There has been a growing push to get rid of life-saving exceptions. In the worldview most folks in the anti-abortion movement have, abortion is murder. It's worse not only in the sense that it's certain death, but that it's intentional. From their standpoint, if some women die because they are refused care, that isn't a certain death, there isn't intentionally going to be a death, so that's the lesser of the evils in that situation" (Stern 2022). Importantly, even in cases where there are life of the mother exceptions, these are very vague, and doctors can still be charged for performing the abortion in emergencies (for instance, in Texas and Oklahoma). This has resulted in the refusal or postponement of care; women have lost their uterus, wound up on breathing machines, and had other severe health outcomes (Tanner 2022).

4. This is a Kantian point. I am assuming that God is a kind of moral person, that is, an intelligent will. On Kant's view of the divine will, see Kain (2021).

5. Furthermore, the idea of double predestination is morally reprehensible, since it implies God creates and predestines some to eternal damnation, presumably for the sake of testing the elect.

6. There are, of course, significant differences between Kant and Plato, although Kant was greatly influenced by Plato. On this see Reich (1939a, 1939b), as well as Baum (2019).

7. Pereboom calls the incompatibilism of this kind of freedom with causal determinism *leeway incompatibilism.* He rightly distinguishes it from *source* incompatibilism. Leeway incompatibilism requires "the ability to do otherwise," whereas source incompatibilism requires that the self "be the undetermined source of one's own actions" Pereboom (2006, p. 542); cf. Pereboom (2001, chps. 1–4). One can hold to source incompatibilism and not to leeway incompatibilism. For instance, Kant and other rationalists held that God is good by necessity and hence cannot choose evil, but is nevertheless *free* since nothing outside of God's nature determines God's activity. So Kant, "...freedom does not consist in the contingency of an action (in its not being determined by any ground at all), i.e., not in indeterminism...but in absolute spontaneity" (Rel. 6:50). I cannot, in the scope of this paper, engage all the contemporary literature on this issue. For a good discussion of some of the main issues at stake, see Fischer et al. (2007).

8. Plantinga's Free-Will Defense relies on the strong understanding of freedom outlined above: the same individual, with the same past up to the moment of choice, and the same causal laws at play, has different possibilities for action at that moment. For instance, he notes that "if God *causes* Curley to go right with respect to A or brings it about that he does so, then Curley isn't free with respect to A" (Plantinga 1974, p. 47). Presumably creating beings that would only freely choose the good, God would have brought it about that they did that (by creating *only* those) so that they would not have *really* been free. Freedom requires the real possibility to choose wrongly. Importantly, Mackie had already anticipated this kind of move, and notes that the idea of freedom envisioned here is incoherent: "If it is replied that this objection is absurd, [the objection that God could have created free beings that always do the good]..., it would seem that 'freedom' must here mean complete randomness or indeterminacy, including randomness with the alternatives good and evil, in other words that people's choices and consequent actions can be 'free' only if they are not determined by their characters . . . . But then if freedom is randomness, how can it be a characteristic of *will*? And still more, how can it be the most important good? What value or merit would there be in free choices if these were random actions which were not determined by the nature of the agent?" (Mackie 1955, p. 209).

9. What I mean by free here is that they chose in accordance with what they most fundamentally wanted, or what their true self wanted.

10. So Nietzsche: "The entire inner world, originally as thin as if it were stretched between two membranes, expanded and extended itself, acquired breadth, depth, and height, in the same measure as the outward discharge was *inhibited*" (Nietzsche [1887] 1995, p. 520).

[11] My approach here is notably different from that of Brian Davies and Mark Murphy. Murphy rejects "God's being bound by any moral norms whatsoever" (Murphy 2021, p. 227). Davies argues that God is not subject to moral obligations, noting that "we have philosophical reasons to deny that God is a moral agent" (Davies 2006, p. 91). Simply denying the premise that God is a moral agent may be one way to deal with the problem of evil, but it comes at great cost. Why should I *hope* in the existence of such a being? Why should I consider it worthy of worship? Without my thinking of this being in moral terms, I must think of it simply as a terrible power and might. If I were to relate to such a being at all, it would be only in terms of fear. On these points, see my forthcoming review of Murphy's book in the *European Journal of Philosophy.* The argument that I am developing in this paper is that *assuming* that we think of God according to moral concepts that we can discover through our reason and that apply to all intelligences, it is still the case that the conditions of God's agency are distinct from our own. There are many things that it would be immoral for us do because of our limited knowledge, power, and goodness; to deny these facts about ourselves would be to deny our creaturely status.

[12] Kant puts this very nicely when he notes: "Although it certainly sounds questionable, it is in no way reprehensible to say that every human being *makes a God* for himself, indeed. he must make one according to moral concepts (attended by the infinitely great properties that belong to the faculty of exhibiting an object in the world commensurate to these concepts) in order to honor in him *the one who made him.* For in whatever manner a being has been made known to him by somebody else, and described as God, indeed, even if such a being might appear to him in person (if this is possible), a human being must yet confront this representation with his ideal first, in order to judge whether he is authorized to hold and revere this being as Divinity. Hence, on the basis of revelation alone, without that concept being *previously* laid down in its purity at its foundation as touchstone, there can be no religion, and all reverence for God would be *idolatry*" (Kant [1793] 1996, Religion, 6:169n).

[13] *What* this is certainly cannot be established outside of a faith tradition. Additionally, even *in* those traditions, the final telos and its realization is a matter of faith and hope—not of knowledge. Further, very few, if any, are in a position to explain what exactly union with God amounts to.

[14] As Adams notes, " . . . Christians never believed that God was a pleasure maximizer anyway" (Adams 1989, p. 298). In fact, I know none of the world's great religions that ever made such a claim, either.

[15] So Adams (1989, p. 306), "philosophical and religious theories differ importantly on what valuables they admit into their ontology." For instance, those who have had mystical experiences report on the *incommensurability* between their experience of God or the holy and their everyday experience of the world. Those who have not had these experiences for themselves may be puzzled by the claims of those who have had them and may be incapable of comprehending those claims. Rudolf Otto, for instance, provides a rich phenomenology of the holy as a category of value. The individual who experiences the holy is struck dumb and is utterly fascinated by the numinous. It is experienced as that which is most ultimately *real* in comparison with which all earthly reality seems like a mere dream. The numinous is experienced as having ultimate value; the creature comes to consider itself "but dust and ashes" in relation to it. While it can be experienced as something terrible, it also instils longings for it that are completely distinct from our "sensuous, psychical, or intellectual impulses and cravings." Our desire for it has its seat in "the highest part of our nature," which mystics called "the basis or ground of the soul" (Otto 1950, p. 36). As Adams (1989, p. 310) notes, agreement on value "is not necessary to consensus on internal consistency." The believer can point to the fact that their belief system remains internally consistent precisely because spiritual values not recognized by the atheologian undergird it.

[16] As an example, take the tenth-century Jewish thinker Saadiah Gaon (1948, pp. 246–47), quoted by Stump (2008, p. 197): "Now He that subjects the soul to its trials is none other than the Master of the universe, who is, of course, acquainted with all its doings. This testing of the soul [that is, the suffering of Job] has been compared to the assaying by means of fire of [lumps of metal] that have been referred to as gold or silver. It is thereby that the true nature of their composition is clearly established. For the original gold and silver remain, while the allows that have been mingled with them are partly burned and partly take flight . . . The pure, clear souls that have been refined are thereupon exacted and enobled."

[17] My own understanding of soul-making has been deeply influenced by (Hick [1977] 2007). However, I differ from him significantly in that my own account rejects as inherently unintelligible an understanding of freedom implying leeway indeterminism.

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
