# Peer review of "Is There a Right to Hope That God Exists? Evil and the Principle of Non-Parity"

_religions, doi:10.3390/rel13100977_

Round 1
Reviewer 1 Report
The author has an interesting paper that I think is worth publishing. The author shows at various points in Sterba's thought that Sterba is thinking of God too anthropomorphically. Moreover, Sterba's conception of freedom is also somewhat dated. Once you acknowledge these things, Sterba's arguments lose force. This should be said.
There are three things I want to mention. First, on a more minor note, the author misinterprets American heartbeat laws to suggest that such laws have no exception for the life of the mother. If he/she has one in mind, I'd cite it. If not, then I think there needs to be some rephrasing in this section.
On a more relevant note, I think the paper could use more sign posting to help the overall structure of the paper.
Finally, I think it would be good to cite from someone like Brian Davies or DBH and how if we assume Classical Theism where God is not a person (at least in a univocal sense) and is not a being (but is Being itself, or as some have put it, beyond being), then the non-parity points become even more obvious.
Reviewer 2 Report
Report on Is There A Right To Hope That God Exists
Lines 64-66—the authors writes that Sterba “the a) equivocal character of political freedom and autonomy, b) the equivocal character of the conditions of agency in God and in creatures implying that there is non-parity in how each must apply the single moral law…” Does the author think that these terms, concepts, or properties are intrinsically equivocal? Or, rather, does the author think that Sterba equivocates on two meanings? Some clarity here would be useful.
Lines 91-4—The author gives a counterexample to Sterba’s first formulation of his “Pauline Principle.” I agree that the principle is not plausible at all. But author’s example—involving abortion and “heartbeat” laws—is overly political. Author should consider an alternative, less political example.
Lines 114—Author mentions ‘this point’ but I’m not sure what specific point they have in mind. Perhaps they have in mind the distinction between means and mere means? If so, they should say so.
Line 116—author writes, “persons have a fundamental value; they are, in fact the ground of all value.” The latter is not equivalent to the former. In fact, the latter is extremely implausible. The author might think about removing it, since it seems unnecessary for their argument.
Lines 124-133—the author formulates three principles. I cannot tell if these are direct quotations or not from Sterba, using just the type setting. Additionally, the third principle doesn’t have a name while the first two do. I am not sure why that is.
Lines 168-170—authors mentions many understandings of “free will” and then switches to “political freedom and autonomy.” Are these all the same? Is author using these terms interchangeably? Author should clarify this terminology.
Lines 185-6—author introduces their own views about freedom. I think this discussion could be improved by engaging more contemporary authors. (For instance, a lot of what they write reminds me of George Sher and T. M. Scanlon.) Author might consider doing more research here.
Lines 235-7—the author considers whether or not a person loves justice for its own sake “by stripping them of all the consequences of justice.” But then the author considers not just stripping them of all of the consequences of justice, but also punishing the person in extreme ways. But that is not a very good test of whether or not someone loves something for its own sake. I might love the taste of a nice IPA for its own sake. But that doesn’t mean I am going to let my eye get gouged out for it. There’s some kind of mismatch here between the examples the author is giving and the point they want to make.
Lines 255-256—author thinks that Sterba’s argument only works because he conflates (or equivocates?) between two different senses of freedom. I think that is probably right. But author should consider adding a passage or quotation from Sterba that illustrates the equivocation.
Lines 318-320—a number of authors have developed the idea that God does not have obligations to us. The most recent is Mark Murphy, in his book God’s Own Ethics. Author should consider consulting this work.
Lines 356-7—author writes that we have “no understanding of the final telos of the soul’s life” But on most monotheistic religion, we do have such an understanding—the understanding is to have union with God. I’m not sure I know what author has in mind here.
Line 420—author writes, when discussing skeptical theism, “Without this knowledge, there is no basis for the kinds of claims Sterba makes.” Again, I think that is probably correct. But I think the author could say more to give evidence that this is true. A quotation or passage or illustration from Sterba would be useful.
